# Effects and Mechanisms of Exosomes from Different Sources in Cerebral Ischemia

**DOI:** 10.3390/cells11223623

**Published:** 2022-11-15

**Authors:** Ruoxi Xie, Xinbing Zeng, Huan Yan, Xiaoping Huang, Changqing Deng

**Affiliations:** 1Hunan Provincial Key Laboratory for Prevention and Treatment of Integrated Traditional Chinese and Western Medicine on Cardio-Cerebral Diseases, Hunan University of Chinese Medicine, Changsha 410208, China; 2Hunan Provincial Key Laboratory of Integrated Traditional Chinese and Western Medicine, Hunan University of Chinese Medicine, Changsha 410208, China

**Keywords:** exosome, cerebral ischemia, neuronal function recovery, intercellular communication, therapeutic effect

## Abstract

Cerebral ischemia refers to the symptom of insufficient blood supply to the brain. Cells of many different origins participate in the process of repairing damage after cerebral ischemia occurs, in which exosomes secreted by the cells play important roles. For their characteristics, such as small molecular weight, low immunogenicity, and the easy penetration of the blood–brain barrier (BBB), exosomes can mediate cell-to-cell communication under pathophysiological conditions. In cerebral ischemia, exosomes can reduce neuronal damage and improve the brain microenvironment by regulating inflammation, mediating pyroptosis, promoting axonal growth, and stimulating vascular remodeling. Therefore, exosomes have an excellent application prospect for the treatment of cerebral ischemia. This article reviews the roles and mechanisms of exosomes from different sources in cerebral ischemia and provides new ideas for the prevention and treatment of cerebral ischemia.

## 1. Introduction

Cerebral ischemia is a life-threatening cerebral vascular disease [1]. As one of the leading causes of morbidity and mortality worldwide, cerebral ischemia is caused by the blockage of blood vessels due to thrombus or embolus [2]. The consequent insufficient blood supply can lead to severe metabolic and cellular symptoms, and eventually results in neuronal cell death and cerebral infarction. Therefore, injuries such as oxidative stress, intracellular calcium overload, excitatory amino acid accumulation, inflammation, blood–brain barrier (BBB) disruption, and peri-infarct depolarization are induced, which further aggravate the death of cells in the central nervous system (CNS) [3]. Meanwhile, along with the occurrence of these lesions, the processes of injury repair and functional recovery are hence triggered, including angiogenesis, neurogenesis, synaptogenesis, and oligodendrogenesis [4].

Extracellular vesicles (EVs) are diverse nanoscale membrane vesicles with diameters ranging from 30 to 1000 nm that are released by cells [5]. They are secreted via the outward budding of the plasma membrane, or the inward budding of the endosomal membrane [6,7]. Based on their biogenesis, size, and biophysical properties, EVs are classified into exosomes, microvesicles, and apoptotic bodies [8]. Exosomes are a subtype of EVs with a diameter of 40~160 nm, originating through the limiting membrane of late endosomes [9,10]. They were first found during an investigation of transferrin receptor fate during the maturation of sheep reticulocytes into red blood cells in 1983 [11,12], and were named as ‘exosomes’ in 1987 [13]. Found in many natural body fluids, such as blood, cerebral spinal fluid (CSF) [14], semen, saliva, plasma, serum, and bronchial fluid [15], exosomes have been shown to be secreted by almost all cell types [16]. In recent years, exosomes have gained extensive attention for their content composition [17]. They have a lipid bilayer structure, with cholesterol, sphingomyelin, ceramide, and other lipids being enriched on their surface [18]. Cell-derived exosomes contain various microRNAs (miRNAs), mRNAs, DNA [16], and proteins [19], which are the major mediators of cell–cell communication, playing indispensable roles in many different physiological processes [20]. The nanoscale sizes and low expression of membrane-bound proteins give exosomes some unique features, such as low immunogenicity, innate stability, and high delivery efficiency [21], which enables exosomes to participate in some important cellular processes, such as proliferation, differentiation, and immune regulation [22]. Furthermore, exosomes can pass across the BBB and take part in suppressing neuroinflammation and promoting neuron generation [23]. These characteristics have bought more and more attention to the therapeutic effects and potential applications of exosomes in the clinical treatment of ischemic stroke [24]. Evidence has shown that exosomes can provide a novel way to alleviate ischemic brain damage through the promotion of angiogenesis [25], suppressing cell apoptosis [26], reducing inflammation [27], and modulating the cell pyroptotic process [23]. Exosomes derived from both the central system and the peripheral system play an essential role in the mitigation [28] and repair of injury caused by ischemia stroke [29]. This paper reviews the effects and mechanisms of exosomes from different sources in cerebral ischemia, which offer a more profound understanding of the role of exosomes underlying ischemic strokes, and contributes to the exploration and development of new diagnostic methods and therapies.

## 2. The Effects and Mechanisms of Exosomes Derived from Central Nervous System Cells in Cerebral Ischemia

Most types of cells in the CNS and peripheral system can secrete exosomes, including microglia, neurons, astrocytes [30], and endothelial cells [31]. The exosomes released from cells in the CNS contribute to cell-to-cell communication and intercellular signaling. Furthermore, they can pass BBB and blood–cerebrospinal fluid barriers, serving as communication between the periphery and the brain [32], which invests exosomes with enormous potential for clinical application.

### 2.1. Neuron-Derived Exosomes

Neural stem cells (NSCs) are a group of ectodermal progenitor cells that can differentiate into committed neural sub-types such as neurons, astrocytes, or oligodendrocytes [33]. Recent studies have shown that neural stem cell-derived exosomes (NSCs-exos) and cortical neuron-derived exosomes play injury-reducing and neuroprotective [34] roles in hypoxic-ischemic brain injury. Research indicates that NSCs-exos exert their neuroprotective effects via the transfer of miR-150-3p into injured cells, which inhibits the CASP2 signaling pathway, thus suppressing neuronal apoptosis and promoting neuron proliferation after brain injury [35]. miR-181c-3p contained in cortical neuron-derived exosomes is found to be an important molecule in the modulation of neuroinflammation after brain ischemia, as it can inhibit the expression of chemokine (C-X-C motif) ligand 1 (CXCL1) and the production of inflammatory factors by astrocytes, which reduces the inflammation burden and achieves the effect of protecting the nerves after ischemic brain injury (IBI). This finding may reveal a therapeutic approach for alleviating the symptoms of IBI through the regulation of astrocyte-induced inflammation [36]. Neural progenitor cell-derived extracellular vesicles (NPC-EVs) were shown to be effective against neuron inflammation, while their poor lesion region-targeting performance raises challenges to their clinical application in the therapy of ischemic stroke. Hence, researchers have managed to combine NPC-EVs with arginine-glycine-aspartic acid (RGD)-4C peptide, which has been proven to be a success for transferring therapeutic molecules to injured areas in the ischemic brain. The experimental results showed that RGD-treated NPC-EVs could improve the targeting of the lesion region of the ischemic brain and suppress poststroke inflammation [26], which leads to a potential therapeutic approach for ischemic stroke. Human induced pluripotent stem cell-derived neural progenitor cells (iPSC-NPCs) have many common points with cortical neurons in morphology and immunohistochemistry. In the microenvironment of ischemic stroke, iPSC-NPCs have a better potential for promoting cellular survival and proliferation, as well as differentiation into mature neurons or astrocytes [37,38]. Studies have shown that, through the regulation of the PTEN/AKT signaling pathway and neurite outgrowth, iPSC-NPC-derived exosomes could protect neurons against damage caused by oxygen–glucose deprivation (OGD) [39].

### 2.2. Microglia-Derived Exosomes

As the unique immune cells of the CNS, microglia play essential roles in many cerebral physiological processes, including regulating immune responses and promoting neuronal survival [40]. Brain damage induced by ischemia stroke can activate microglia and polarize them into either the “classically activated” M1 phenotype or the “alternatively activated” M2 phenotype [41]. Plenty of evidence suggests that M1 microglia promote the release of pro-inflammatory cytokines, inhibit synaptic growth, and aggravate acute brain damage [16]. On the contrary, the M2 phenotype is identified as a “healing” phenotype that establishes neuroprotective effects on the brain and improves long-term neurological outcomes after stroke [42]. Various kinds of miRNAs contained in microglia-derived exosomes are involved in the process of alleviating the inflammatory response, reducing damage, and promoting repair after ischemic hypoxic brain injury. Through the secretion of exosomes containing miRNA-26a, interleukin-4-polarized microglia cells can promote angiogenesis, thus alleviating ischemic stroke-induced damage [43]. Exosomes secreted by microglia in the M2 phenotype (BV2-exos) can transport the exosomal miRNA-137 targeting gene Notch1 to attenuate neuronal apoptosis, which contributes to the amelioration of ischemia–reperfusion (I/R) brain injury [44]. BV2-exos can moderate neuronal apoptosis after stroke occurs and promote the survival of neurons via the secretion of miR-124 enriched in exosomes and its downstream target USP14 [16]. Furthermore, a group of researchers have found that the neuroprotective effect of Vinpocetine could be associated with their actions on microglia cells. By inhibiting phosphodiesterase enzyme1-B (PDE1-B) in microglial cells, Vinpocetine could not only inhibit the polarization of the M1 microglial phenotype, but also enhance autophagic flux, which is associated with the alteration of exosomal contents and properties for protecting the survival and neurite structures of neurons against ischemic stroke [45]. Nevertheless, the effects of microglia-derived exosomes may not always be positive in the hypoxia state. Fibroblast growth factor2 (FGF2) can promote endothelial cell angiogenesis by activating STAT3, whereas upregulated miR-424-5p in microglial exosomes in hypoxia can inhibit the activation of the STAT3 pathway by targeting FGF2, thus resulting in brain microvascular endothelial cell (BMEC) injury. Therefore, the inhibition of miR-424-5p may promote endothelial cell angiogenesis and reduce neuronal damage [46].

### 2.3. Astrocyte-Derived Exosomes

The astrocyte is the largest type of glial cell. Astrocytes are widely distributed throughout the CNS. Their unique morphology can provide structural support to neurons. Furthermore, astrocytes have other functions, such as regulating neuroinflammatory responses, modulating synaptic activity, supplying energy to neurons, and maintaining the blood–brain barrier (BBB) [47]. Astrocytes also play a part in regulating autophagy level after brain injury occurs. Numerous experimental results have shown that insufficient or excessive levels of autophagy could aggravate cell death, while modest autophagy could protect cells against stressful circumstances and facilitate cell survival [48]. In vivo and in vitro experiments have confirmed that astrocyte-derived exosomes (ASC-exos) could ameliorate ischemia-induced neuronal damage via the inhibition of OGD-induced neuron autophagy [49]. By studying exosomes isolated from an OGD-based ischemia model in vitro, researchers have found that circular RNA circSHOC2 in ischemic-preconditioned astrocyte-derived exosomes suppress neuronal apoptosis and ameliorate cellular damage through the regulation of autophagy and its action on miR-7670-3p, thereby upregulating sirtuin1 (SIRT1) levels. The findings of this research may lead to a novel therapy for ischemic stroke treatment [50]. Through the suppression of histone demethylase KDM6B, downregulating bone morphogenetic proteins (BMP2), and silencing si-Bcl-2 modifying factor (BMF), astrocyte extracellular vesicle (ASC-EV)-derived miR-22-3p can significantly decrease apoptosis, mitigate the I/R brain injury in vivo, and enhance neuron viability in vitro [51]. Another study has suggested that ASC-exos that carry miR-17-5p could improve neurobehaviors, reduce neuronal apoptosis and cerebral infarction, and inhibit oxidative stress responses, as well as inflammation, in vivo and in vitro [52]. Except the miRNAs mentioned above, some other substances in astrocyte exosomes have also attracted attention. ASC-exos with prostaglandin D2 (PGD2) synthase expression can promote axonal outgrowth and functional recovery after stroke [53]. Astrocytes can also be linked to the mechanism of ischemic preconditioning (IPC). Neurons can take up exosomes released by OGD-preconditioned astrocytes. In this process, miR-92b-3p contained in exosomes is transferred from preconditioned astrocytes to neurons, which contributes to the attenuation of OGD-induced neuron death and apoptosis and protects neurons against OGD damage [54]. Another study demonstrated that, when using in vitro exosome treatment via delivering gap junction alpha 1 (GJA1), exosomes released by astrocytes can be taken up by neurons, leading to the downregulation of the apoptosis level and the upregulation of mitochondrial performance, which promotes the functional recovery of damaged neurons [55].

### 2.4. Brain Microvascular Endothelial Cell-Derived Exosomes

Exosomes derived from cerebral endothelial cells play essential roles in protecting neurons under a hypoxia state. Endothelial cell-derived exosomes (EC-exos) can protect neurons against I/R injury through the promotion of cell growth, migration, and invasion, and the inhibition of the apoptosis of SH-SY5Y nerve cells [56]. MiRNA-126-3p from EC-exos can protect PC12 cells against apoptosis and promote neurite outgrowth. This finding may serve as a therapeutic strategy for mitigating nerve damage and promoting function recovery [57]. Exosomes derived from cerebral endothelial cells (CECs) can enhance axonal growth by altering miRNAs and their target protein profiles, such as RhoA, in recipient neurons. The research also demonstrated that exosomes released by CECs from ischemic stroke-affected rats could promote axonal growth better than CEC-exos from a normal brain [58]. Vascular endothelial cell-derived exosomes (VEC-exos) can promote the survival viability of neural progenitor cells (NPC) and protect NSCs against I/R injury through the facilitation of neuron proliferation, migration, and the inhibition of apoptosis in vitro [59]. Endothelial progenitor cell-derived exosomes (EPC-exos) may alleviate ischemic injury by inhibiting apoptosis and promoting angiogenesis [60]. 

Based on above experiments and researches, it can be concluded that exosomes from CNS alleviate damage after cerebral ischemia and promote recovery through reducing inflammation, suppressing apoptosis, as well as enhancing neurogenesis and angiogenesis (Table 1). 

## 3. The Effects and Mechanisms of Exosomes Derived from Peripheral Cells in Cerebral Ischemia

### 3.1. Mesenchymal Stem-Cell-Derived Exosomes

Mesenchymal stem cells, considered as “sentinel and safe-guards of injury” [61], have been proven to be able to release neurotrophic factors, including glial-derived neurotrophic factor, brain-derived neurotrophic factor, and nerve growth factor, which have shown their therapeutic potential in damage repair [32].

#### 3.1.1. Bone Marrow Mesenchymal Stem-Cell-Derived Exosomes

Bone mesenchymal stem-cell-derived exosomes (BMSC) are a group of unique stem cells with great differential potential. The protective effect of BMSC-derived exosomes (BMSC-exos) is found to be linked to the inhibition of apoptosis in different studies. Various miRNAs can function in an ischemia-hypoxic brain environment taking BMSC-exos as carriers. BMSC-exo-miR-26a-5p mimics can dramatically reduce protein kinase CDK6 levels in BV-2 cells after oxygen–glucose deprivation/reoxygenation (OGD/R) treatment, as well as in brain tissues of the middle cerebral artery occlusion–reperfusion (MCAO/R) model. The downregulation of CDK6 mediated by exosomal miR-26a-5p might contribute to the attenuation of I/R injury in vivo, and the inhibition of microglia apoptosis in vitro [62]. By downregulating the gene PTEN, and therefore activating the PI3K/Akt/mTOR pathway, the miR-17-92 cluster enriched can increase corticospinal tract (CST) neuronal plasticity, axonal myelination, and axonal extension under the situation of rat cerebral stroke, which enhances the electrophysiological response and promotes neurological functional recovery in the poststroke phase [63]. Furthermore, BMSC-exos can facilitate angiogenic function and the cellular survival of the hypoxia/reoxygenation (H/R)-affected endothelial cell through the release of miR-126. The mechanism may be associated with the activation of the PI3K/Akt/eNOS pathway, which decreases cleaved caspase-3 expression, and promotes angiogenesis and the generation of growth factors [64]. Another study showed that the PI3K/Akt/eNOS pathway can also be activated by miR-132-3p-enriched BMSC-exos, thus mitigating H/R-induced oxidative stress and endothelial cell apoptosis [65]. By targeting tumor necrosis factor receptor associated factor 6 (TRAF6), microRNA-124-3p contained in BMSC-exos improves neurological functions, alleviates neuron pathological and structural damage, suppresses oxidative stress, and reduces neuronal apoptosis in newborn hypoxic-ischemic brain damage (HIBD) rats [66]. BMSC-exos carrying miR-455-3p can attenuate hippocampal neuronal injury in MCAO/R mice and OGD/R-induced N2a cell injury by increasing N2a cell activity and decreasing apoptosis through miR-455-3p targeting PDCD7 [67]. BMSC-exos can also inhibit apoptosis by targeting TGR5 [68]. miR-150-5p from BMSCs can enhance the therapeutic effects of BMSC-exos on cerebral I/R injury via a reduction in B-cell translocation gene 2 (TLR5) expression in MCAO rats, which contributes to a decrease in inflammatory factor levels, the inhibition of neuron apoptosis, the mitigation of pathological change, and an improvement in neurological function [69]. These studies provide a potential therapeutic strategy for the treatment of cerebral infarction.

Apart from the miRNA mentioned above, BMSC-exos are also found to function in protecting neurons through other mechanisms and pathways. It has been proven that the nucleotide-binding domain and leucine-rich repeat family protein 3 (NLRP3) inflammasome plays an essential role in neuronal damage induced by I/R [70]. BMSC-exos can promote microglial polarization toward M2, which suppresses NLRP3 inflammasome-mediated inflammation and pyroptosis, thus alleviating cerebral I/R injury [71]. Another study demonstrated that BMSC-exos could ameliorate NLRP3 inflammasome-mediated pyroptosis by promoting AMP-activated kinase (AMPK) dependent autophagic flux [72]. In addition, exosomes derived from CXC motif chemokine receptor type 4 (CXCR4) overexpressing BMSCs exhibit antiapoptotic effect via the Wnt-3a/β-catenin pathway, which facilitates the proliferation and tube formation of microvascular endothelial cells in cerebral I/R injury [73]. Researchers also noticed that cysteinyl leukotrienes (CysLTs), as potent inflammatory mediators, were largely produced with the decomposition of necrotic cells, while BMSC-exos could reverse CysLT2R-ERK1/2’s effect of inducing M1 microglia polarization, and promote the differentiation of microglia to the M2 phenotype, thus attenuating brain injury and ameliorating microglial inflammation meditated by M1 microglia [74]. An experiment intravenously administered BMSC-derived small extracellular vesicles (sEVs) obtained from bone marrow sample of a healthy donor (2 × 10^6^ or 2 × 10^7^ BMSC equivalents/kg) to young and aged mice. The results showed that sEVs at both doses promoted periinfarct angiogenesis in both types of rats. In addition, low-dose sEVs enhanced neurogenesis in the subventricular zone [75]. Additionally, BMSC-derived small extracellular vesicles (BMSC-sEVs) can decrease the infiltrates of inflammatory cells, such as leukocytes, monocytes, polymorphonuclear neutrophils, and macrophages, in the brains of aged mice with ischemic stroke. In peripheral blood, the number of monocytes and activated T cells are remarkably decreased by BMSC-sEVs, further reducing inflammation and mediating postischemic neuroprotection in brains of aged mice [76]. Molecular imaging technique showed that in the ischemic brains of mice, BMSC-exos treatment could significantly facilitate angiogenesis and neurogenesis and reduce the expression of IL-1β as well [77]. This research also showed that BMSC-exos could migrate into the brains of mice with ischemic stroke, which provides a new approach of clinical remission or therapy of ischemic stroke [77]. BMSC-exos treatment can attenuate OGD/R-induced oxidative stress and the dysregulation of mitochondrial function-associated genes in hippocampal neurons, which makes BMSC-exos treatment a potential therapeutic strategy to prevent neuronal damage induced by OGD/R [78]. Buyang Huanwu Decoction (BYHWD) treatment can promote angiogenetic miRNA and vascular endothelial growth factor (VEGF) expression in BMSC-exos, thus upregulating angiogenesis in the rat brain [79]. Another group of researchers observed that iron oxide nanoparticles (IONP) stimulated the expression of therapeutic growth factors in the MSC. They found that magnetic extracellular nanovesicles (MNV) derived from IONP-harboring BMSCs, supported by magnetic navigation, could improve the ischemic-lesion localization and the therapeutic effects of BMSC-exos, which enhances their abilities in anti-inflammation and anti-apoptosis, and promotes angiogenesis in ischemic brain lesions, thus contributing to a significant decrease in infarction volume and to the amelioration of motor function. The MNV injection may be applied to avoid the major defect of current BMSC-exos treatment or nanovesicles (NV) treatment in ischemic stroke [80].

#### 3.1.2. Adipose-Derived Mesenchymal Stem Cells

Pigment epithelium-derived factor (PEDF) is a multifunctional protein that has neurotrophic, anti-inflammatory, and neuroprotective functions. Researchers have found that exosomes released by adipose-derived mesenchymal stem cells (ADSCs-exos) with increased PEDF content could further protect neurons against OGD-induced apoptosis in cerebral I/R through the activation of autophagy and the suppression of neuronal apoptosis. It is also observed that blocking autophagy could lead to a reduction in the effect of PEDF-containing exosomes [81]. Exosomes from hypoxic pre-treated ADSCs can alleviate brain injury caused by acute ischemic stroke via the delivery of circ-Rps5. Circ-Rps5 can induce the overexpression of its downstream targets miR-124-3p or the downregulation of SIRT7, which promotes the polarization of microglia from the M1 to M2 phenotype under lipopolysaccharide (LPS) conditions [82]. Another study showed that the systemic administration of ADSCs-exos remarkably promoted the expression of von Willebrand factor, an endothelia cell marker, and doublecortin, a neuroblasts marker, which reduced neuron cell death and enhanced cell proliferation in comparison with the control group. Moreover, researchers found that exosomes from miRNA-126-modified ADSCs could inhibit microglial activation and the expression of inflammatory factors in vivo and in vitro, thus improving functional recovery, inhibiting neuroinflammation, and enhancing neurogenesis. The result of this study suggested that the intravenous administration of miR-126^+^ exosomes in poststroke might represent a novel treatment for stroke [83].

#### 3.1.3. Human Umbilical Cord Mesenchymal Stem Cell-Derived Exosomes

Multiple studies have shown that human umbilical cord mesenchymal stem cell-derived exosomes (hUMSC-exos) can reduce microglia-mediated neuroinflammation, protecting against brain injury. Exosomes derived from umbilical cord mesenchymal stem cells can interfere with the Toll-like receptor 4 (TLR4) signaling of BV-2 microglia, which contributes to reducing neuroinflammation induced by microglia in perinatal brain injury [84]. HUMSC-exos carrying miR-146a-5p can alleviate the neuroinflammatory response mediated by microglial via the suppression of the IRAK1/TRAF6 pathway [85]. Furthermore, hUMSC-exos can enhance mitophagy and alleviate subsequent neuronal injury by increasing FOXO3a expression, therefore attenuating OGD/R-induced microglial pyroptosis [86]. Researchers also explored the clinical application potential of hUMSC-exos. A study showed that hUMSC-exos injected into rats with deep vein thrombosis (DVT) could upregulate the delivery of miR-342-3p, which downregulated the expression of endothelin A receptor and eventually alleviated DVT. The above findings may provide a novel clinical treatment for DVT [87].

### 3.2. Plasma and Serum Exosomes

Research indicated that plasma-derived exosomes (PLA-exos) enriched with heat shock protein 70 (HSP70) showed better brain targeting and treatment effects. PLA-exos with inherited HSP70 can facilitate the migration and diapedesis of PLA-exos through an interaction with endothelial TLR4. Furthermore, PLA-exos-meditated HSP70 delivery can activate tight junction protein (TJP), which leads to the inhibition of mitochondria-meditated neuron cell apoptosis, the suppression of ROS accumulation, and the alleviation of BBB damage [88]. Edaravone-loaded PLA-exos could mitigate ischemic damage in brain tissue via the interaction between transferrin on the surface of PLA-exos and transferrin receptor (TfR) on the surface of brain endothelial cells, which reduces ROS generation [89]. Researchers have also found that melatonin-treated exosomes effectively reduced the infarct volume and improved function recovery via the regulation of the TLR4/Nuclear Factor kappa B (NF-kB) signaling pathway, thus further enhancing therapeutic effects of PLA-exos against inflammatory responses induced by ischemic stroke and inflammasome-mediated pyroptosis [90]. Another study showed that growth arrest and DNA damage-inducible protein 34 (GADD34) level were increased in PLA-exos of cerebral ischemic rats, which might be the consequence of the dephosphorylation of eukaryotic translation initiation factor 2α (eIF2α) and the phosphorylation of p53. Furthermore, GADD34 inhibitor treatment inhibited neuronal apoptosis, decreased the infarct volume, and improved functional outcomes in cortical penumbra after ischemic stroke [91]. Additionally, a study revealed that circulating plasma exosomes were related to the increased risk of stroke after varicella zoster virus reactivation caused herpes zoster (HZ, shingles). Compared to exosomes from patients without HZ, HZ exosomes can initiate platelets to form platelet–leukocyte aggregates and contain the proteins that can concert recipient cells to a prothrombotic state. Therefore, the application of antiplatelet agents for HZ may be a feasible clinical practice to decrease stroke risk [92].

Exosomes from remote ischemic preconditioning (RIPC) serum show neuroprotective effects via the upregulation of miRNA-126, thus reducing the expression of DNMTs3B in neurons and raising OGD tolerance in SH-SY5Y cells [93]. However, research has also proven that serum exosomal miR-27-3p could target PPARγ to stimulate the activation of microglia and the expression of inflammatory cytokines, thus aggravating acute cerebral infarction (ACI). Nevertheless, the root cause of upregulated exosomal miR-27-3p expression in serum after ACI requires further study [94]. Moyamoya disease (MMD) is a rare steno-occlusive and slowly progressing cerebrovascular disorder, and its pathogenesis mechanism is yet unknown. Researchers have found that the proliferation levels of mouse brain vascular EC cells were significantly increased, and that more ethynyl-2-deoxyuridine-positive cells were generated after treatment with MMD serum-derived exosomes (SDEs). However, it was also observed that SDEs from an ischemic MMD patient promoted neuroblastoma cell proliferation, and SDEs from hemorrhagic MMD patients induced the dysfunction of the mitochondria in cerebrovascular ECs. The finding may help with understanding the pathogenesis mechanisms of MMD and provide new therapeutic strategies for the disease [95]. 

According to the related studies we can summarize that the effects of exosomes from peripheral nervous system in brain ischemic stroke are mainly exerted through facilitating brain tissues functional recovery, inflammation reduction, as well as inhibition of cells pyroptosis and apoptosis (Table 2).

## 4. The Effects and Mechanisms of Exosomes Derived from Other Sources in Cerebral Ischemia

Apart from the above-mentioned sources, macrophages and some other cells can also react to brain injury via exosomes.

Through the regulation of microglial polarity from phenotype M1 to anti-inflammatory phenotype M2, LPS-stimulated macrophage-derived exosomes have functional improvements and neuroprotection effects after ischemic stroke [96]. Macrophage-derived exosomes can serve as carriers of various drug molecules to enhance or to facilitate the functions of drugs towards targeting pathways or cells. Exosomes from heptapeptide-loaded (Hep-loaded) macrophages can reduce mitochondrial injury in astrocytes via the suppression of dynamin-related protein-1 (Drp1)-fission 1 (Fis1) interaction after I/R, alleviating mitochondria-mediated neuronal damage [97]. Edaravone-loaded macrophage-derived exosomes enhance the targeting performance of Edaravone toward ischemic lesion areas in rat brains with permanent middle cerebral artery occlusion (PMCAO), which further reduces neuronal cell death and promotes the polarization of microglia from M1 to M2 [98]. Curcumin-laden exosomes from macrophages can alleviate cerebral I/R injury by downregulating ROS accumulation in lesions, thus reducing BBB damage and suppressing neuronal apoptosis mediated by mitochondria [99]. Another research team isolated exosomes from the supernatants of interleukin-4-induced M2-polarized macrophages, and discovered that M2-macrophages-derived exosomes can activate the nuclear factor erythroid related factor 2 (Nrf2)/heme-oxygenase-1 (HO-1) signaling pathway, which contributes to inhibiting the generation of ROS and malondialdehyde, reducing the release of lactate dehydrogenase, increasing cell activity, and eventually achieving the purpose of protecting HT22 neurons [100].

Human umbilical endothelial cells (HUECs) can reduce the apoptosis of neurons under OGD in a HUEC-neuron coculture assay. Researchers also found that Cav-1 upregulated by neurons during ischemia stroke can increase the neuron intake of extracellular vesicles derived from an endothelial cell, which attenuates apoptosis via exosomal miR-1290 and protects neurons [101]. Exosome release via stem cell-derived dental pulp can mitigate cerebral I/R damage by inhibiting the inflammatory response mediated by the HMGB1/TLR4/MyD88/NF-κB pathway [102]. Exosomes released by human urine-derived stem cells (USCs) can enhance neuronal differentiation and the proliferation of NSCs after OGD/R via the exosomal miR-26a/histone deacetylase 6 (HDAC6) axis [103]. Human USCs can also release exosomes enriched with miR-21-5p, which promotes early nerve formation by regulating the Eph receptor A(EPha4)/tyrosine kinase (TEK) axis [104]. EVs from human hypoxic olfactory mucosa MSCs (OM-MSCs) were proven to be able to promote the migration, proliferation, and angiogenic activities of human brain microvascular endothelial cells via the exosomal miR-612–TP53–hypoxia-inducible factor 1-alpha (HIF-1α)–vascular endothelial growth factor (VEGF) axis [105]. Another team performed an OGD-affected IEC-6-primary cortical neuron coculture system under normothermia (37 °C) and therapeutic hypothermia (TH) (32 °C) conditions. The researchers found that I/R-injured intestinal epithelium cells can induce cortical neuron death via the production of paracrine mediators such as exosomal miRNAs associated with necroptosis, apoptosis, and/or pyroptosis, whereas TH can counteract this process and protect cortical neurons in stroke patients [106]. Except for the exosomes from animal cells, researchers also isolated and characterized novel plant exosome-like nanoparticles (ELNs) from Momordica charantia (MC) and found that MC-ELNs attenuate ischemia–reperfusion-induced damage to the BBB and suppress neuronal apoptosis, probably via the modulation of the phosphoinositide-3-kinase (PI3Ks)/protein kinase B (AKT)/glycogen synthase kinase (GSK3β) signaling pathway [107]. Moreover, based on in vivo and in vitro experiments, researchers discovered the effects of extracted MC-exosomes (MCEs) in the inhibition of platelet activation, and the aggregation, adhesion, and platelet-mediated migration of HCT116 cells, which may reveal the potential of MCEs in therapies for stroke and tumor metastasis [108]. 

Therefore, we can conclude that exosomes from other sources also have protective effects in cerebral ischemic stroke, and possess great potential as therapy of brain ischemia (Table 3).

## 5. Conclusions and Prospects

It can be concluded from the above review that exosomes derived from diverse sources targeting at different brain cells (Figure 1) through different mechanism (Figure 2) result in corresponding protective (mainly) or damaging effects on the brain tissues after ischemia. However, the current studies are mostly limited to a single substance in exosomes from a single source, and whether there are synergistic effects of multiple substances in exosomes to function in the process of cerebral ischemia needs further exploration because the interactions between nerve cells are complex. Moreover, how do brain parenchymal cells or remote organs affect the secretion of exosomes and the changes of their internal substances through cell signaling pathways? How do exosomes regulate the expression of endogenous genes in recipient cells? These questions are waiting to be answered. Secondly, as molecular markers for disease diagnosis, exosomes have great potential in regulating neurological function recovery after ischemic stroke. Nevertheless, there is still room for improvement in the purity and quality of the extraction, although the method of extracting exosomes has gradually matured. Efficient and reliable exosome isolation technologies still require more study and investigation. Furthermore, exosomes can combine with drugs as carriers to play a synergistic role. There are many studies focusing on the combination of exosomes and drugs in the mitigation and treatment of ischemic stroke [26,45,53,80,89,90,97,98,99,105]. In addition, the association between exosomes and the formation of thrombus has been confirmed, as well as the accumulation of blood platelets, which may provide new ideas in the treatment of thrombolysis, thrombectomy, and anti-platelet in cerebral stroke. However, large-scale clinical trials are needed to verify their clinical feasibility. The potential adverse effects of exosomes applied in clinical therapy are as yet blurry. Improving targeting through the modification of exosomes to make them easier to reach the ischemic lesion is also a key direction for carrier-exosome therapy research, which is expected to provide strong support for the clinical treatment of cerebral ischemia. Overall, the clinical application of exosomes in the treatment of cerebral ischemia is a novel and promising therapeutic approach. The study of the mechanism of exosomes acting in the process of cerebral ischemia is of great significance for guiding the applications of exosomes in clinical treatment.

## Figures and Tables

**Figure 1 cells-11-03623-f001:**
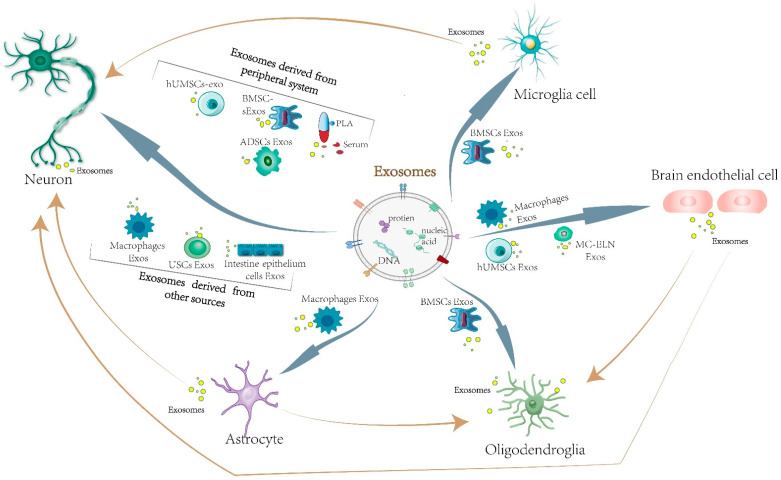
Target cells that exosomes from different sources influence in ischemic brain tissues. After ischemic stroke, exosomes can be released from cells of the CNS (neural stem cells, microglia, astrocyte, vascular endothelial cells), peripheral system, and other sources to intervene neurocyte. Exosomes secreted by peripheral mesenchymal stem cells can act on microglia. Macrophages, human umbilical cord mesenchymal stem cells, and momordica charantia exosomes can alleviate the injury of brain endothelial cells and protect the BBB. Astrocyte, brain endothelial cells and mesenchymal stem cells target at oligodendroglia, promoting remyelination. Moreover, exosomes derived from macrophages cells can affect astrocyte.

**Figure 2 cells-11-03623-f002:**
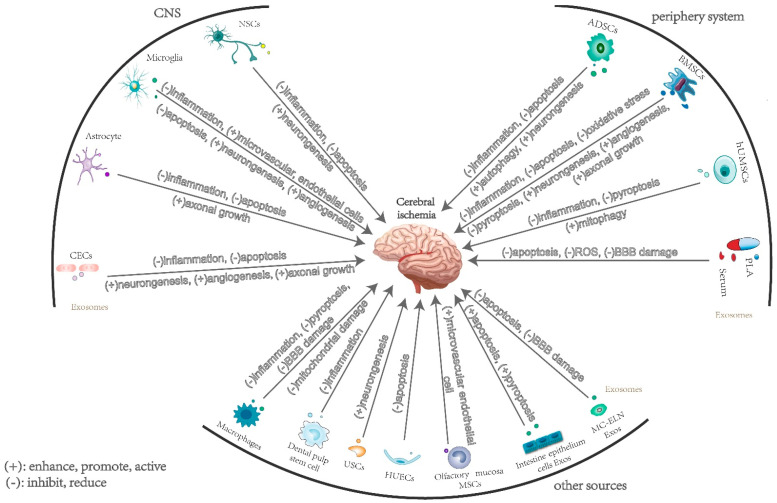
Mechanism of exosomes from different sources exert on ischemia brain tissues. The exosomes acting on cerebral ischemia mainly come from cells in the CNS and peripheral system. Exosomes from CNS can mitigate nerve cell damage through inhibition of neuronal apoptosis. Moreover, NSC-exos can promote neuronal proliferation in cerebral ischemia; microglia-derived exosomes have the effect of meditating inflammatory responses, as well as promoting neurogenesis and angiogenesis. ASC-exos function by inducing autophagy and regulating inflammation; exosomes from brain endothelial cells play a crucial role in protecting neurons from hypoxic injury, alleviating apoptosis, and promoting axonal growth. Exosomes from mesenchymal stem cells, including BMSCs, ADSCs, and hUMSCs, have the functions of reducing inflammation, mediating autophagy, regulating cell pyroptosis, promoting angiogenesis, and protecting neurons. Moreover, the researchers found that serum and plasma-derived exosomes, macrophage-derived exosomes, and exosomes of some other sources were also actively involved in the physiological regulation process after cerebral ischemia.

**Table 1 cells-11-03623-t001:** Sources and components of central nervous system cell-derived exosomes, and their mechanism and functions.

Source of Exosomes	Component	Mechanism	Function	Experiment Type	Reference
NSCs		Transferred into ischemic cells, and repair tissue	Reduce injury and protect neurons	In vitro	[34]
NSCs	miR-150-3p	Inhibit CASP2 signaling pathway	Anti-apoptosis, promote neurogenesis	In vitro and in vivo	[35]
Cortical neuron	miR-181c-3p	Inhibit the expression CXCL1	Anti-inflammation	In vitro	[36]
NPC-EVs			Anti-inflammation, suppressing cell apoptosis	In vitro and in vivo	[26]
iPSC-NPCs		PTEN/AKT signaling pathway and neurite outgrowth	Promote neurogenesis	In vitro	[39]
Microglia		Inhibition of PDE1-D	Anti-inflammatory, enhance autophagic flux	In vitro and in vivo	[45]
Microglia	miR-424-5p	Inhibit STAT3 pathway	Injure microvascular endothelial cells	In vitro and in vivo	[46]
M2 microglia	miR-124	Target USP14	Attenuate neuronal apoptosis and promote neurogenesis	In vitro	[16]
M2 microglia	miR-137	Target gene Notch1	Attenuate neuronal apoptosis	In vitro and in vivo	[44]
M2 microglia	miR-26a		Promote angiogenesis	In vitro and in vivo	[43]
Astrocyte			Inhibit autophagy	In vitro and in vivo	[49]
Astrocyte	circular RNA circSHOC2	Regulate autophagy and miR-7670-3p/SIRT1	Suppress neuronal apoptosis	In vitro and in vivo	[50]
Astrocyte	miR-22-3p	Suppress KDM6B-mediated effects on the BMP2/BMF axis	Enhance neuron viability, anti-apoptosis	In vitro and in vivo	[51]
Astrocyte	miR-17-5p		Anti-apoptosis, anti-oxidation, anti-inflammation	In vitro and in vivo	[52]
Astrocytes			Axonal neuronal outgrowth	In vitro and in vivo	[53]
Astrocytes	miR-92b-3p		Attenuated neuron death and apoptosis	In vitro	[54]
Astrocyte		Deliver GJA1	Anti-apoptosis	In vitro and in vivo	[55]
ECs			Anti-apoptosis and promote neurogenesis	In vitro and in vivo	[56]
ECs	miR-126-3p		Anti-apoptosis and increase neurite outgrowth	In vivo	[57]
CECs	miRNAs	Target RhoA	Facilitate axonal growth	In vitro	[58]
VECs			Anti-apoptosis and promote neurogenesis	In vitro and in vivo	[59]
EPCs			Inhibit apoptosis and promote angiogenesis	In vivo	[60]

NSCs, neural stem cells; NPC-EVs, neural progenitor cell-derived extracellular vesicles; iPSC-NPCs, human induced pluripotent stem cell-derived neural progenitor cells; ECs, endothelial cells; CECs, cerebral endothelial cells; VECs, vascular endothelial cells; EPCs, endothelial progenitor cells.

**Table 2 cells-11-03623-t002:** Sources and components of peripheral system cell-derived exosomes, their mechanisms, and their functions.

Source of Exosomes	Component	Mechanism	Function	Experiment Type	Reference
BMSCs	miR-26a-5p	Downregulate CDK6	Inhibit microglia apoptosis, reduce inflammation	In vitro and in vivo	[62]
BMSCs	miR-17-92	Downregulate gene PTEN, activate the PI3K/Akt/mTOR pathway	Increase axonal extension and axonal myelination	In vivo	[63]
BMSCs	miR-126	PI3K/Akt/eNOS pathway	Promote angiogenesis, downregulate caspase-3 of ECs	In vitro	[64]
BMSCs	miR-132-3p	PI3K/Akt/eNOS pathway	Ameliorate endothelial apoptosis and oxidative stress	In vitro and in vivo	[65]
BMSCs	miR-124-3p	Target TRAF6	Suppress oxidative stress and reduce neuronal apoptosis	In vivo	[66]
BMSCs	miR-455-3p	Target PDCD7	Anti-apoptosis	In vitro and in vivo	[67]
BMSCs		Target TGR5	Anti-apoptosis	In vitro and in vivo	[68]
BMSCs	miR-150-5p	Reduce TLR5 expression	Repress inflammation, block neuron apoptosis	In vivo	[69]
BMSCs		promote microglial polarization toward M2	Inhibit NLRP3 inflammasome-mediated inflammation and pyroptosis	In vitro and in vivo	[71]
BMSCs		Promote AMPK-dependent autophagic flux	Ameliorate NLRP3 inflammasome-mediated pyroptosis	In vitro	[72]
BMSCs		Wnt-3a/β-catenin pathway	Anti-apoptotic, facilitate the proliferation and tube formation of MSCs	In vitro and in vivo	[73]
BMSCs		Promote the differentiation of microglia to the M2	Reverse CysLT2R-ERK1/2′s effect, reduce inflammation	In vitro and in vivo	[74]
BMSCs			Increase peri-infarct angiogenesis	In vivo	[75]
BMSCs		Decrease the infiltrates of inflammatory cells	Anti-inflammation	In vivo	[76]
BMSCs		Reduce the expression of IL-1β	Facilitate angiogenesis and neurogenesis	In vivo	[77]
BMSCs			Alleviate oxidative stress and dysregulation of mitochondrial function	In vitro	[78]
BMSCs	miRNA and VEGF		Promote angiogenesis	In vitro and in vivo	[79]
BMSCs			Inhibit inflammation and apoptosis, as well as promote angiogenesis	In vivo	[80]
ADSCs		Increase PEDF content	Activate autophagy and suppress neuronal apoptosis	In vitro and in vivo	[81]
ADSCs	circ-Rps5	MiR-124-3p overexpression or SIRT7 downregulation	Attenuate inflammation	In vitro and in vivo	[82]
ADSCs	miR-126		Enhance neurogenesis and inhibit neuroinflammation	In vitro and in vivo	[83]
hUMSCs	miR-146a-5p	Suppress IRAK1/TRAF6 pathway	Anti-neuroinflammation	In vitro and in vivo	[85]
hUMSCs		Toll-like receptor 4 signaling of BV-2 microglia	Reduce microglia-mediated neuroinflammation	In vitro and in vivo	[84]
hUMSCs		Increase FOXO3a expression	Enhance mitophagy, attenuate pyroptosis	In vitro	[86]
hUMSCs		Upregulate miR-342-3p and downregulate endothelin A receptor expression	Alleviate DVT	In vitro and in vivo	[87]
PLA	HSP70		Reduces ROS, apoptosis, and BBB damage	In vitro	[88]
PLA		Interaction between transferrin and transferrin receptor	Reduces ROS generation	In vivo	[89]
PLA		TLR4/NF-κB signaling pathway	Enhance plasma exosome against inflammatory responses and pyroptosis	In vivo	[90]
PLA		Dephosphorylation of eIF2α and phosphorylation of p53	Induce neuronal apoptosis	In vivo	[91]
HZ PLA			Form platelet-leukocyte aggregates	Human	[92]
RIPC serum	miR-126	Downregulate DNMTs3B	Reduce SH-SY5Y cells injure	In vivo	[93]
Serum exosomes	miR-27-3p	Target PPARγ	Promote inflammation, thereby aggravating ACI	In vitro and in vivo	[94]
MMD serum			Promote neuroblastoma cells proliferation	In vivo	[95]

ADSCs, adipose-derived mesenchymal stem cells; BMSCs, bone mesenchymal stem cells; hUMSCs, human umbilical cord mesenchymal stem cells; DVT, deep vein thrombosis; HZ, herpes zoster; PLA, plasma; RIPC, remote ischemic preconditioning; ACI, aggravating acute cerebral infarction; MMD, moyamoya disease.

**Table 3 cells-11-03623-t003:** Sources and components of other cell-derived exosomes, their mechanisms, and their functions.

Source of Exosomes	Component	Mechanism	Function	Experiment Type	Reference
Macrophages		Modulating microglial polarity	Anti-inflammation	In vitro and in vivo	[96]
Macrophages		Drp1-Fis1 interaction	Reduce mitochondrial damage in astrocytes	In vitro and in vivo	[97]
Macrophage		Targets neuronal cells and microglia	Reduce inflammation response	In vivo	[98]
Macrophages		Downregulating ROS	Protect BBB and antagonize neuronal apoptosis	In vitro and in vivo	[99]
M2 macrophages		Activate Nrf2/HO-1 signaling pathway	Inhibit ROS, protect HT22 neurons	In vitro	[100]
Dental pulp stem cell		Inhibit HMGB1/TLR4/MyD88/NF-κB pathway	Anti-inflammation	In vitro and in vivo	[102]
USCs		MiR-26a/HDAC6 axis	Promote neurogenesis	In vitro and in vivo	[103]
USCs	miR-21-5p	EPha4/TEK axis	Promote neurogenesis	In vitro and in vivo	[104]
Olfactory mucosa MSCs	miR-612-TP53-HIF-1α-VEGF axis		Promote the formation of HBMECs	In vitro and in vivo	[105]
HUECs	miR-1290	Cav-1 upregulates intake of HUECs-EVs	Attenuates apoptosis	In vitro and in vivo	[101]
Intestinal epithelium cells	miRNA		Promote apoptosis, necroptosis, and/or pyroptosis of cortical neurons	In vitro	[106]
MC-ELNs		AKT/GSK3β signaling pathway	Protect BBB and anti-apoptosis	In vivo	[107]
MCEs			Inhibit platelet activation, aggregation, adhesion, and HCT116 cells migration	In vitro and in vivo	[108]

USCs, urine-derived stem cells; MSCs, mesenchymal stem cells; HUECs, human umbilical endothelial cell; MC, momordica charantia; ELNs, exosome-like nanoparticles; MCEs, MC-exosomes.

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
