# Peer review of "Effects and Mechanisms of Exosomes from Different Sources in Cerebral Ischemia"

_cells, 2022, doi:10.3390/cells11223623_

Round 1

Reviewer 1 Report

Ischemic stroke is one of the leading causes of disability and death worldwide. Despite the occurrence of new therapies (thrombolysis, thrombectomy), the burden of stroke is rising and the need for innovative therapies is one of the major clues for neurosciences. Indeed, Xie et al. purpose a large and relevant review about the relationships between exosomes and cerebral ischemia. The paper reports the roles and mechanisms of exosomes implied in cerebral ischemia from various sources central or peripheric cells with an exhaustive and well-written review. They provide a solid amount of evidence about the key rule of exosomes in pathophysiology of cerebral ischemia.

Several minor revisions could emphasize the reach of the review

1.     The authors purpose a long list of effects of exosome in cerebral ischemia from different type of cells. The description could be more relevant if a dynamic modeling of brain ischemia at the cellular level with the different roles of exosome was added

2.     In addition, the authors could add a figure with the different locations of secretion of exosomes and their potential rule to counteract ischemia induced brain damages

3.     The authors did not provide data or hypothesis related to the potential interactions between effects of stroke therapies (thrombolysis, thrombectomy, antiplatelet medications) and exosomes. If these data are available, the manuscript would be more clinically relevant for neurologists.

4.     Is there some available data about exosome and ischemia of others organ, for example in myocardial infarction?

5.     In table 1, authors could add a column to precise the type of paper where the rule of exosome was assessed (in vitro, in vivo (animal or human?))

Reviewer 2 Report

1.      Authors must define exosomes, extracellular vesicles in the first paragraph. They have redundantly used these terms.  

2.      Providing original article reference for key statements are essential. For example, reference 5 doesn’t hold good for the statement.

3. There is no delimitation of role of exogenous versus endogenous exosomes in action.

  4. New insights in terms of future prospects are weakly written

Round 2

Reviewer 2 Report

1.      Split the first paragraph into two or three. Second paragraph can start with extracellular vesicles from line 35.

2.      In the line 58, remove the word “accumulating” as you have provided information from a one simple paper.

3.      In the exosomes derived from peripheral cells section, exosomes from MSCs were discussed in first section. Authors should specify the source of MSCs in each of these cases. For example exosomes from mouse bone marrow derived MSCs, etc.  
